# Extensive impact of low-frequency variants on the phenotypic landscape at population-scale

Téo Fournier, Omar Abou Saada, Jing Hou, Jackson Peter, Elodie Caudal, Joseph Schacherer*

Université de Strasbourg, CNRS, GMGM UMR 7156, Strasbourg, France

**Abstract** Genome-wide association studies (GWAS) allow to dissect complex traits and map genetic variants, which often explain relatively little of the heritability. One potential reason is the preponderance of undetected low-frequency variants. To increase their allele frequency and assess their phenotypic impact in a population, we generated a diallel panel of 3025 yeast hybrids, derived from pairwise crosses between natural isolates and examined a large number of traits. Parental versus hybrid regression analysis showed that while most phenotypic variance is explained by additivity, a third is governed by non-additive effects, with complete dominance having a key role. By performing GWAS on the diallel panel, we found that associated variants with low frequency in the initial population are overrepresented and explain a fraction of the phenotypic variance as well as an effect size similar to common variants. Overall, we highlighted the relevance of low-frequency variants on the phenotypic variation.

DOI: https://doi.org/10.7554/eLife.49258.001

## Introduction

Natural populations are characterized by an astonishing phenotypic diversity. Variation observed among individuals of the same species represents a powerful raw material to develop better insight into the relationship existing between genetic variants and complex traits (*Mackay et al., 2009*). The recent advances in high-throughput sequencing and phenotyping technologies greatly enhance the ability to determine the genetic basis of traits in various organisms (*Alonso-Blanco et al., 2016*; *Auton et al., 2015*; *Mackay et al., 2012*; *Peter et al., 2018*). Dissection of the genetic mechanisms underlying natural phenotypic diversity is within easy reach when using classical mapping approaches such as linkage analysis and genome-wide association studies (GWAS) (*Mackay et al., 2009*; *Visscher et al., 2017*). Alongside these major advances, however, it must be noted that there are some limitations. All genotype-phenotype correlation studies in humans and other model eukaryotes have identified causal loci in GWAS explaining relatively little of the observed phenotypic variance of most complex traits (*Eichler et al., 2010*; *Hindorff et al., 2009*; *Manolio et al., 2009*; *Shi et al., 2016*; *Stahl et al., 2012*; *Wood et al., 2014*; *Zuk et al., 2014*).

Despite the efforts made to find the genetic variants responsible for complex traits, the variants found explain only a small part of the heritability, that is of the fraction of the phenotypic variance explained by the underlying genetic variability. One of the most striking examples is observed with human height. This trait is estimated to be 60–80% heritable (*Speed et al., 2017*; *Visscher et al., 2008*) but close to 700 variants found in an analysis based on more than 250,000 individuals only explain 20% of this total heritability (*Wood et al., 2014*). Multiple justifications for this so-called missing heritability have been suggested, including the presence of low-frequency variants, (*Gibson, 2012*; *Hindorff et al., 2009*; *Manolio et al., 2009*; *Pritchard, 2001*; *Walter et al., 2015*),

*For correspondence:
schacherer@unistra.fr

Competing interests: The authors declare that no competing interests exist.

structural variants (*e.g.* copy number variants) (*Peter et al., 2018*), small effect variants, as well as the low power to estimate non-additive effects (*Cordell, 2009*; *Mackay, 2014*; *Zuk et al., 2012*).

Variants present in less than 5% of the individuals are coined as low-frequency variants and are known to be involved in a large number of rare Mendelian disorders (*Gibson, 2012*). However, implication of rare variants is also pervasive in common diseases and other complex traits. Assessing the impact and effect of low-frequency variants at a population scale and on a large phenotypic spectrum will allow to gain better insight into the genetic architecture of the phenotypic variation in a species. As GWAS cannot deal with low-frequency and rare variants due to statistical limitations, except for very large sample sizes, their effect has often been overlooked.

Among model organisms, the budding yeast *Saccharomyces cerevisiae* is especially well suited to dissect variations observed across natural populations (*Fay, 2013*; *Peter and Schacherer, 2016*). *S. cerevisiae* isolates can be found in a broad array of biotopes both human-associated (*e.g.* wine, sake, beer and other fermented beverages, food, human body) or wild (*e.g.* plants, soil, insects) and are distributed world-wide (*Peter et al., 2018*). Phenotypic diversity among yeast isolates is significant and the *S. cerevisiae* species presents a high level of genetic diversity ($\pi = 3 \times 10^{-3}$), much greater than that found in humans (*Lek et al., 2016*). Because of their small and compact genomes, an unprecedented number of 1,011 *S. cerevisiae* natural isolates has recently been sequenced (*Peter et al., 2018*). Yeast genome-wide association analyses have revealed functional Single Nucleotide Polymorphisms (SNPs), explaining a small fraction of the phenotypic variance (*Peter et al., 2018*). However, these analyses highlighted the importance of the copy number variants (CNVs), which account for a larger proportion of the phenotypic variance and have greater effects on phenotypes compared to the SNPs. Nevertheless, even when CNVs and SNPs are taken together, the phenotypic variance explained is still low (approximately 17% on average) and consequently a large part of it is unexplained.

Interestingly, much of the detected genetic polymorphisms in the 1011 yeast genomes dataset are low-frequency variants with almost 92.7% of the polymorphic sites associated with a minor allele frequency (MAF) lower than 0.05. This trend is similar to that observed in the human population (*Auton et al., 2015*; *Walter et al., 2015*) and definitely raised a question regarding the impact of low-frequency variants on the phenotypic landscape within a population and on the missing heritability (*Zuk et al., 2014*). Here, we investigated the underlying genetic architecture of phenotypic variation as well as unraveling part of the missing heritability by accounting for low-frequency genetic variants at a population-wide scale and non-additive effects controlled by a single locus. For this purpose, we generated and examined a large set of traits in 3025 hybrids, derived from pairwise crosses between a subset of natural isolates from the 1,011 *S. cerevisiae* population. This diallel crossing scheme allowed us to capture the fraction of the phenotypic variance controlled by both additive and non-additive phenomena as well as infer the main modes of inheritance for each trait. We also took advantage of the intrinsic power of this diallel design to perform GWAS and assess the role of the low-frequency variants on complex traits.

## Results

### Diallel panel and phenotypic landscape

Based on the genomic and phenotypic data from the 1,011 *S. cerevisiae* isolate collection (*Peter et al., 2018*), we selected a subset of 55 isolates that were diploid, homozygous, genetically diverse (*Figure 1a*), and originated from a broad range of ecological sources (*Figure 1b*) (*e.g.* tree exudates, *Drosophila*, fruits, fermentation processes, clinical isolates) as well as geographical origins (Europe, America, Africa and Asia) (*Figure 1c* and *Supplementary file 1*). A full diallel cross panel was constructed by systematically crossing the 55 selected isolates in a pairwise manner (*Figure 1d*). In total, we generated 3025 hybrids, representing 2970 heterozygous hybrids with a unique parental combination and 55 homozygous hybrids. All 3025 hybrids were viable, indicating no dominant lethal interactions existed between the parental isolates. We then screened the entire set of the parental isolates and hybrids for quantification of mitotic growth abilities across 49 conditions that induce various physiological and cellular responses (*Figure 1—figure supplement 1*, *Figure 1—figure supplement 2*, *Supplementary file 2*). We used growth as a proxy for fitness traits (see

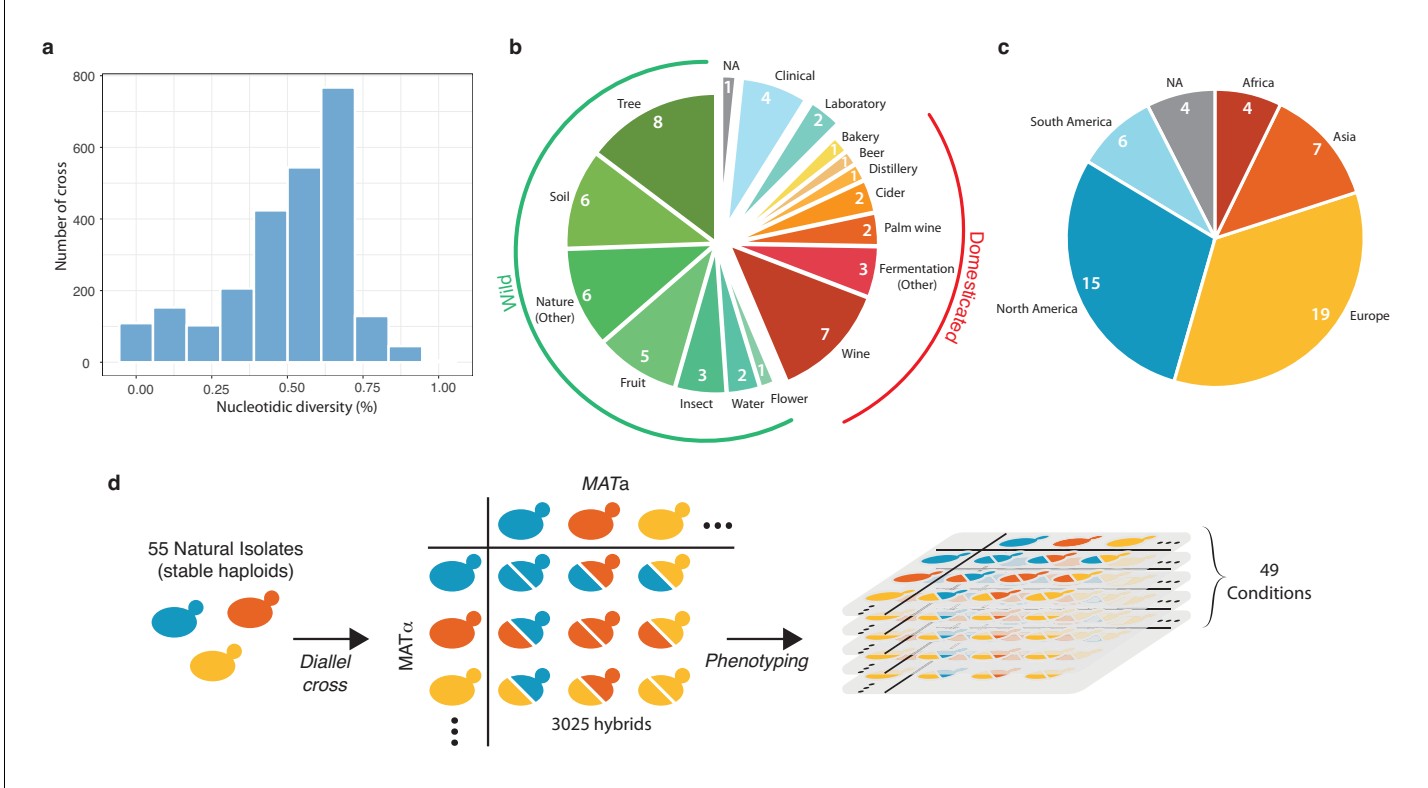

**Figure 1.** Diversity of the 55 selected natural isolates and diallel design. (**a**) Pairwise sequence diversity between each pair of parental strains. (**b**) Ecological origins of the selected strains. See also *Supplementary file 1*. (**c**) Geographical origins of the selected strains. (**d**) Generation of the diallel hybrid panel. 55 natural isolates available as both mating types as stable haploids were crossed in a pairwise manner to obtain 3025 hybrids. This panel was then phenotyped on 49 growth conditions impacting various cellular processes.

DOI: https://doi.org/10.7554/eLife.49258.002

The following source data and figure supplements are available for figure 1:

**Source data 1.** Growth ratios for every hybrid and parental isolate on each growth condition.

DOI: https://doi.org/10.7554/eLife.49258.006

**Figure supplement 1.** Phenotypic variance in hybrids.

DOI: https://doi.org/10.7554/eLife.49258.003

**Figure supplement 2.** Correlation between conditions.

DOI: https://doi.org/10.7554/eLife.49258.004

**Figure supplement 3.** Phenotypic correlation between *MAT*a and *MAT*α isolate.

DOI: https://doi.org/10.7554/eLife.49258.005

Materials and methods). Ultimately, this phenotyping step led to the characterization of 148,225 hybrid/trait combinations.

## Estimation of genetic variance components using the diallel panel (additive vs. non-additive)

The diallel cross design allows for the estimation of additive *vs.* non-additive genetic components contributing to the variation in each trait by calculating the combining abilities following Griffing's model (*Griffing, 1956*). For each trait, the General Combining Ability (GCA) for a given parent refers to the average fitness contribution of this parental isolate across all of its corresponding hybrid combinations, whereas the Specific Combining Ability (SCA) corresponds to the residual variation unaccounted for from the sum of GCAs from the parental combination. Consequently, the phenotype of a given hybrid can be formulated as $\mu + GCA_{parent1} + GCA_{parent2} + SCA_{hybrid}$, where $\mu$ is the mean fitness of the population for a given trait. We found a near perfect correlation (Pearson's r = 0.995, p-value<2.2e-16) between expected and observed phenotypic values, confirming the accuracy of

the model used (see Materials and methods). Using GCA and SCA values, we estimated both broad- ($H^2$) and narrow-sense ($h^2$) heritabilities for each trait (**Figure 1**). Broad-sense heritability is the fraction of phenotypic variance explained by genetic contribution. In a diallel cross, the total genetic variance is equal to the sum of the GCA variance of both parents and the SCA variance in each condition. Narrow-sense heritability refers to the fraction of phenotypic variance that can be explained only by additive effects and corresponds to the variance of the GCA in each condition (**Figure 2a**). The $H^2$ values for each condition ranged from 0.64 to 0.98, with the lowest value observed for fluconazole (1 µg.ml$^{-1}$) and the highest for sodium meta-arsenite (2.5 mM), respectively. The additive part ($h^2$ values) ranged from 0.12 to 0.86, with the lowest value for fluconazole (1 µg.ml$^{-1}$) and the highest for sodium meta-arsenite (2.5 mM), respectively. While broad- and narrow-sense heritabilities are variable across conditions, we also observed that on average, most of the phenotypic variance can be explained by additive effects (mean $h^2$ = 0.55). However, non-additive components contribute significantly to some traits, explaining on average one third of the phenotypic variance observed (mean $H^2$ - $h^2$ = 0.29) (**Figure 2b**). Despite a good correlation between broad- and narrow-sense heritabilities (Pearson's r = 0.809, p-value=1.921e-12) (**Figure 2c**), some

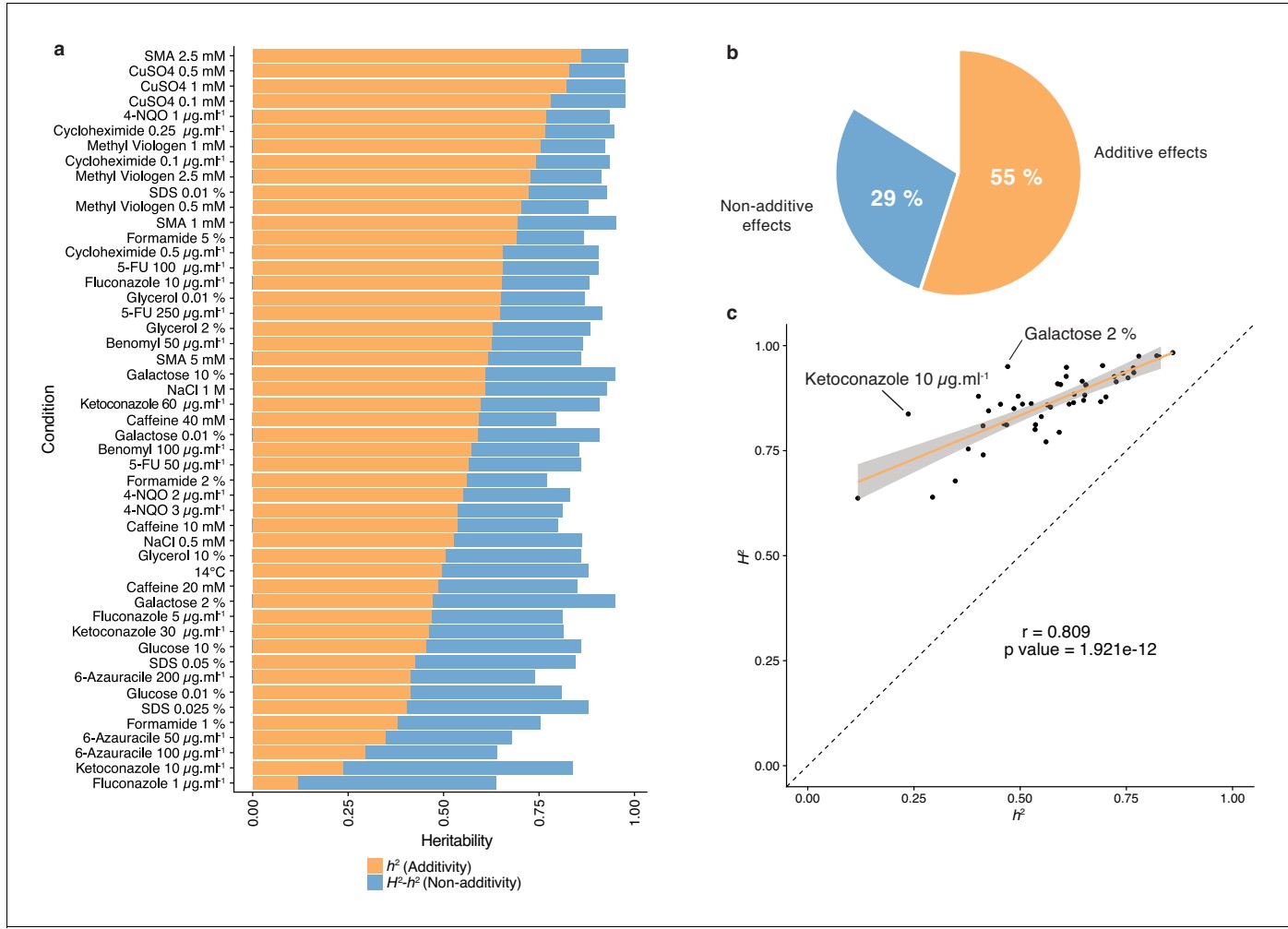

**Figure 2.** Heritability measurements. (**a**) The whole bar represents the overall heritability ($H^2$) for each condition tested. Orange part of the bars represents the narrow-sense heritability $h^2$, that is the fraction of phenotypic variance explained by additive effects, while blue part depicts the fraction of phenotypic variance explained by non-additive effects. (**b**) Overall mean additive and non-additive effects for every tested growth condition. (**c**) Representation of $H^2$ as a function of $h^2$ showing the relative additive versus non-additive effects for each condition. Outlier conditions in terms of non-additive variance will lie further away from the linear regression line. Person's r (95% confidence interval: 0.684–0.889) with the corresponding p-value is displayed.

DOI: https://doi.org/10.7554/eLife.49258.007

traits display a larger non-additive contribution, such as in galactose (2%) or ketoconazole (10 μg/ml). Interestingly, we revealed that these two conditions revealed to be mainly controlled by dominance (see below). Altogether, our results highlight the main role of additive effects in shaping complex traits at a population-scale and clearly show that this is not restricted to the single yeast cross where this trend was first observed (*Bloom et al., 2013*; *Bloom et al., 2015*). Nonetheless, non-additive effects still explain a third of the observed phenotypic variance. This result also corroborates at a species-wide level the extensive impact of non-additive effects on phenotypic variance (*Forsberg et al., 2017*; *Yadav et al., 2016*).

## Relevance of dominance for non-additive effects

To have a precise view of the non-additive components, the mode of inheritance and the relevance of dominance for genetic variance, we focused on the deviation of the hybrid phenotypes from the expected value under a full additive model. Under this model, the hybrid phenotype is expected to be equal to the mean between the two parental phenotypes, hereinafter referred as Mean Parental Value or Mid-Parent Value (MPV). Deviation from this MPV allowed us to infer the respective mode of inheritance for each hybrid/condition combination (*Lippman and Zamir, 2007*), that is additivity, partial or complete dominance towards one or the other parent and finally overdominance or underdominance (*Figure 3a–b*, see Materials and methods). Only 17.4% of all hybrid/condition combinations showed enough phenotypic separation between the parents and the corresponding hybrid, allowing the complete partitioning in the seven above-mentioned modes of inheritance. For the 82.6% remaining cases, only a separation of overdominance and underdominance can be achieved

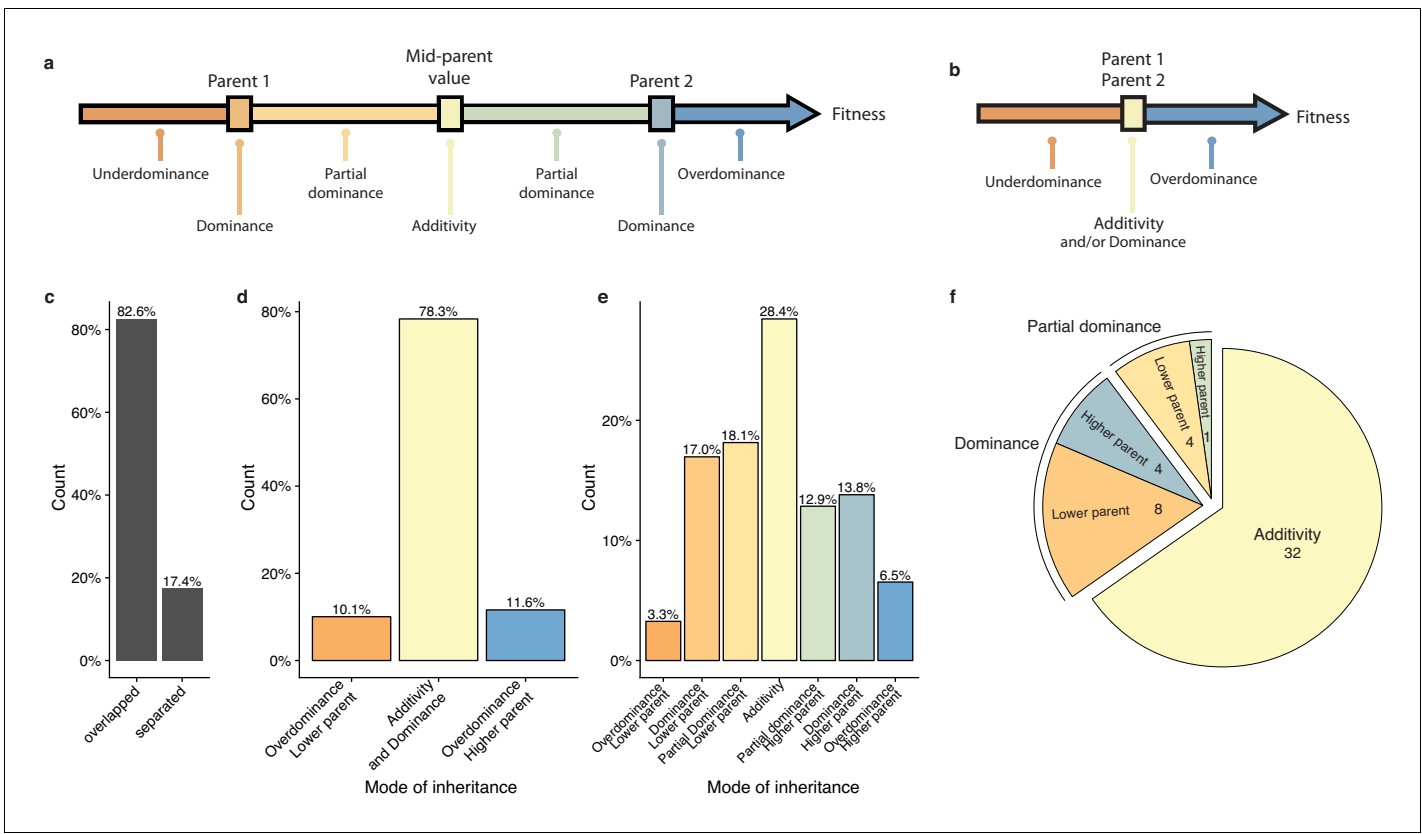

**Figure 3.** Mode of inheritance. (a) Representation of the different mode of inheritance depending on the hybrid value when a separation can be achieved between parental strains and (b) if a clear separation cannot be achieved between parental strains. (c) Percentage of parental phenotypes separated from each other for which a complete partition of different inheritance modes can be achieved. (d) Inheritance modes for every cross and condition where no separation can be achieved between the two homozygous parents. e. Inheritance modes for every cross and condition where a clear phenotypic separation can be achieved between the two homozygous parents. (f) The number of conditions in each main inheritance mode.
DOI: https://doi.org/10.7554/eLife.49258.008

(*Figure 3c*). Interestingly, these events are not as rare as previously described (*Zörgö et al., 2012*), with 11.6% of overdominance and 10.1% of underdominance (*Figure 3d*). When a clear separation is possible (*Figure 3e*), one third of the condition/cross combinations detected were purely additive whereas the rest displayed a deviation towards one of the two parents, with no bias (*Figure 3e*). When looking at the inheritance mode in each condition, most of the studied growth conditions (32 out of 49) showed a prevalence of additive effects (*Figure 3f*). However, 17 conditions were not predominantly additive throughout the population. Indeed, a total of 12 conditions were detected as mostly dominant with 4 cases of best parent dominance, including galactose (2%) and ketoconazole (10 µg.ml$^{-1}$), and 8 of worst parent dominance. The remaining five conditions displayed a majority of partial dominance (*Figure 3f*). These results confirm the importance of additivity in the global architecture of traits, but more importantly, they clearly demonstrate the major role of dominance as a driver for non-additive effects. Nevertheless, the presence of conditions with a high proportion of partial dominance combined with the cases of over and underdominance may indicate a strong and pervasive impact of epistasis on phenotypic variation.

## Diallel design allows mapping of low-frequency variants in the population using GWAS

Next, we explored the contribution of low-frequency genetic variants (MAF <0.05) to the observed phenotypic variation in our population. Genetic variants considered by GWAS must have a relatively high frequency in the population to be detectable, usually over 0.05 for relatively small datasets (*Visscher et al., 2017*). Consequently, low-frequency variants are evicted from classical GWAS. However, the diallel crossing scheme stands as a powerful design to assess the phenotypic impact of low-frequency variants present in the initial population as each parental genome is presented several times, creating haplotype mixing across the matrix and preserving the detection power in GWAS.

To avoid issues due to population structure, we selected a subset of hybrids from 34 unrelated isolates in the original panel to perform GWAS (see Materials and methods, *Supplementary file 1*). By combining known parental genomes, we constructed 595 hybrid genotypes *in* silico, matching one half matrix of the diallel plus the 34 homozygous diploids. We built a matrix of genetic variants for this panel and filtered SNPs to only retain biallelic variants with no missing calls. In addition, due to the small number of unique parental genotypes, extensive long-distance linkage disequilibrium was also removed (see Materials and methods), leaving a total of 31,632 polymorphic sites in the diallel population. Overall, 3.8% (a total of 1,180 SNPs) had a MAF lower than 0.05 in the initial population of the 1,011 *S. cerevisiae* isolates but surpassed this threshold in the diallel panel, reaching a MAF of 0.32 (*Figure 4a–b*).

To map additive as well as non-additive variants impacting phenotypic variation, we performed GWA using two different models (*Seymour et al., 2016*) (see Materials and methods). We used a classical additive model, encoding for SNPs where linear relationship between trait and genotype is assessed, that is every locus has a different encoding for each genotype. To account for non-additive inheritance, we also used an overdominant model, which only considers differences between heterozygous and homozygous thus revealing overdominant and dominant effects. For each of these two models, we performed mixed-model association analysis of the 49 growth conditions with FaST-LMM (*Lippert et al., 2011*; *Widmer et al., 2015*). Overall, GWAS revealed 1723 significantly associated SNPs (*Figure 4—source data 1*) by detecting from 2 to 103 significant SNPs by condition, with an average of 39 SNPs per condition. Minor allele frequencies of the significantly associated SNPs were determined in the 1011 sequenced genomes, from which the diallel parents were selected (*Figure 4*). Interestingly, 16.3% of the significant SNPs (281 in total) corresponded to low-frequency variants (MAF <0.05), with 19.5% of them (55 SNPs) being rare variants (MAF <0.01). This trend is the same and maintained for both models, with 19.3% and 15.2% of low-frequency variants for the additive and overdominant models, respectively. Due to the scheme used, it is important to note that it is possible to increase the MAF of low-frequency variants at a detectable threshold in the diallel panel and to query their effects but it is still difficult for truly rare variants (MAF <0.01), probably leading to an underestimation. However, these results clearly show that low-frequency variants indeed play a significant part in the phenotypic variance at the population-scale. We then estimated the contribution of the significant variants to total phenotypic variation (see Materials and methods) in our panel and found that detected SNPs could explained 15% to 32% of the variance, with a median of 20% (*Figure 4d*). When looking at the variance explained by each variant over their

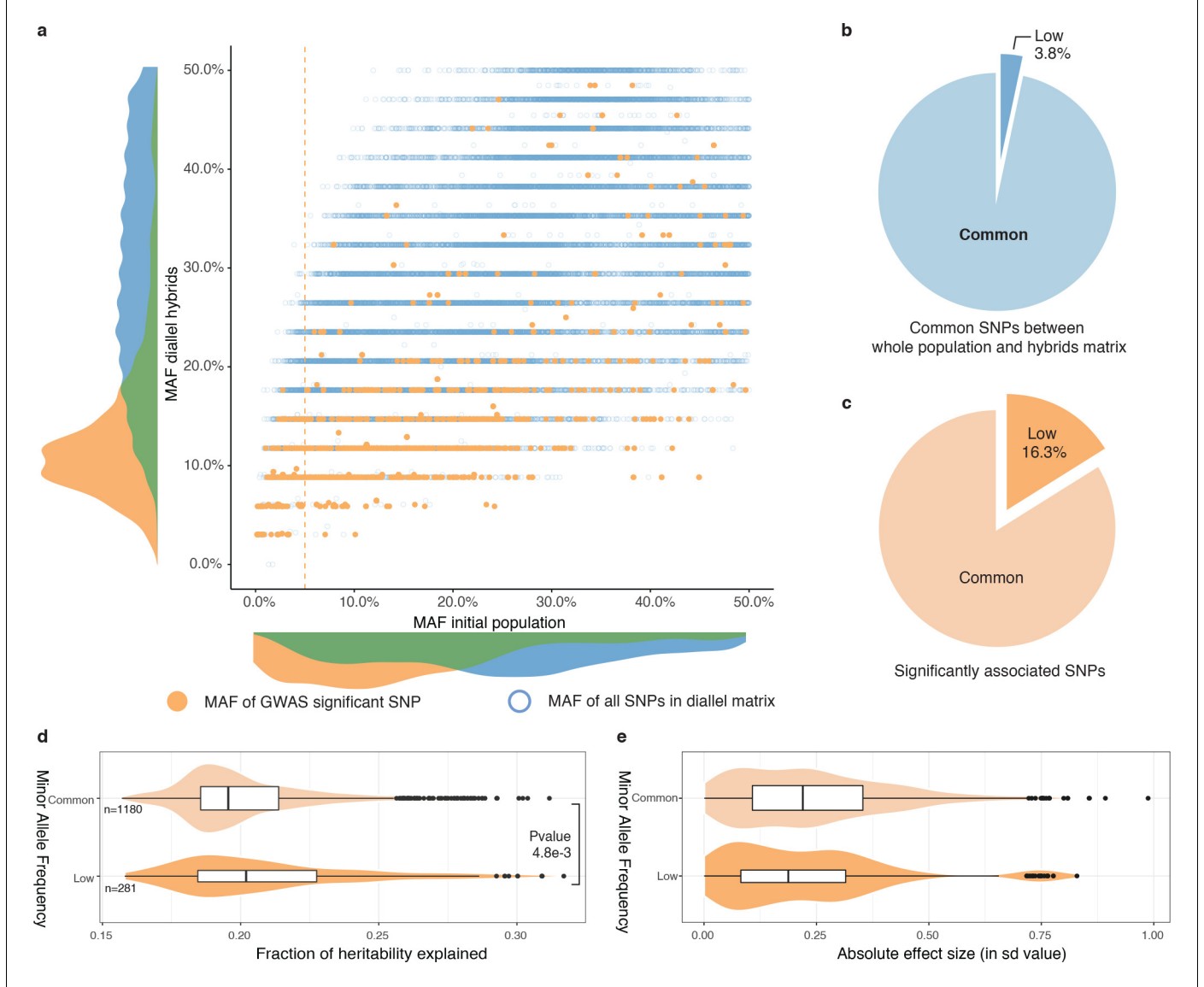

**Figure 4.** Rare and low-frequency variants detection. (**a**) Comparison of MAF for each SNP between the whole population (1011 strains) and the hybrid diallel matrix used for GWAS. Hollow blue circles represent the MAF of all SNPs common to the initial population and the diallel hybrids (31,632). Full orange circles show the MAF of significantly associated SNPs. Vertical orange line shows the 5% MAF threshold. (**b**) Proportion of SNPs with a MAF below 0.05. (**c**) Proportion of significantly associated SNPs with a MAF below 0.05. (**d**) Fraction of heritability explained for common and low-frequency variants. P-value was calculated using a two-sided Mann-Whitney-Wilcoxon test, difference in location of $-4.5e^{-3}$ (95% confidence interval $-7.9e^{-3}$ - $1.4e^{-3}$). (**e**) Absolute effect size of common and low-frequency variants.

DOI: https://doi.org/10.7554/eLife.49258.009

The following source data and figure supplement are available for figure 4:

**Source data 1.** Significantly associated SNPs SNPs without MAF are SNPs that were not biallelic in the initial population of 1011 isolates (*Peter et al., 2018*).
DOI: https://doi.org/10.7554/eLife.49258.011
**Figure supplement 1.** Significantly associated SNPs.
DOI: https://doi.org/10.7554/eLife.49258.010

respective allele frequency, it is noteworthy that low-frequency variants explained roughly the same proportion of the phenotypic variation (median of 20.2%) than the common SNPs (median of 19.6%) (*Figure 4d*). In addition, the variance explained by the associated rare variants were also higher on average than the rest of the detected SNPs (*Figure 4—figure supplement 1a*). It is noteworthy that

this trend was robust and conserved across the two encoding models implemented, accounting for additive and overdominant effects (*Figure 4—figure supplement 1a*). However, these results cannot be extrapolated to the whole population and only hold in the scope of our diallel population where these variants are now overrepresented compared to the natural population. Indeed, variance explained is related to the surveyed population because its value relies on the MAF of the variants. Therefore, in the whole natural population of 1011 isolates, their contribution to the phenotypic variance will be less important because of their lower MAF. To obtain a value that is unrelated to the studied population, we measured their respective effect size (*Figure 4e*). Here again we found that on average, low-frequency variant have about the same effect size (mean of 0.23 sd) than the common variants (mean of 0.25 sd).

To gain insight into the biological relevance of the set of associated SNPs, we first examined their distribution across the genome and found that 62.5% of them are in coding regions (with coding regions representing a total of 72.9% of the *S. cerevisiae* genome) (*Figure 4—figure supplement 1b*), with all of these SNPs distributed over a set of 546 genes. Over the last decade, an impressive number of quantitative trait locus (QTL) mapping experiments were performed on a myriad of phenotypes in yeast leading to the identification of 145 quantitative trait genes (QTG) (*Peltier et al., 2019*) and we found that 19 of the genes we detected are included in this list (*Figure 4—figure supplement 1c*). In addition, 22 associated genes were also found as overlapping with a recent large-scale linkage mapping survey in yeast (*Bloom et al., 2019*) (*Figure 4—figure supplement 1c*). We then asked whether the associated genes were enriched for specific gene ontology (GO) categories (*Supplementary file 3*). This analysis revealed an enrichment (p-value=$5.39 \times 10^{-5}$) in genes involved in 'response to stimulus' and 'response to stress', which is in line with the different tested conditions leading to various physiological and cellular responses.

### *SGD1* and the mapping of a low-frequency variant

Finally, we focused on one of the most strongly associated genetic variant out of the 281 low-frequency variants significantly associated with a phenotype. The chosen variant was characterized by two adjacent SNPs in the *SGD1* gene and was detected in 6-azauracile (100 µg.ml$^{-1}$) with a p-value of 2.75e-8 with the overdominant encoding and 6.26e-5 with the additive encoding. Their MAF in the initial population is only 2.5% and reached 9% in the diallel panel with three genetically distant strains carrying it (*Figure 5a*). The SNPs are in the coding sequence of *SGD1*, an essential gene encoding a nuclear protein. The minor allele (AA) induces a synonymous change (TT**G** (Leu) → TT**A** (Leu)) for the first position and a non-synonymous mutation (**G**AA (Glu)→ **A**AA (Lys)) for the second position (*Figure 5a*). The phenotypic advantage conferred by this allele was observed with a significant difference between the homozygous for the minor allele, heterozygous and homozygous for the major allele (*Figure 5b*). To functionally validate the phenotypic effect of this low-frequency variant, CRISPR-Cas9 genome-editing was used in the three strains carrying the minor allele (AA) in order to switch it to the major allele (GG) and assess its phenotypic impact. Both mating types have been assessed for each strain. When phenotyping the wildtype strains containing the minor allele and the mutated strains with the major allele, we observed that the minor allele confers a phenotypic advantage of 0.2 in growth ratio compared to the major allele (*Figure 5c*) therefore validating the important phenotypic impact of this low-frequency variant. However, no assumptions can be made regarding the exact effect of this allele at the protein-level because no precise characterization has ever been carried out on Sgd1p and no particular domain has been highlighted.

## Discussion

Understanding the source of the missing heritability is essential to precisely address and dissect the genetic architecture of complex traits. Over the years, the diallel hybrid panel design has proven its strength to dissect part of the genetic architecture of traits in populations. One of the main advantages of using such experimental design is the ability to precisely isolate the part of phenotypic variance that is controlled by additive effects from the one controlled by non-additive effects. While our analysis revealed that an important part of the phenotypic variance is linked to additive effects, about a third remains ruled by non-additive interactions encompassing dominance and epistasis. These results are in line with previous findings.

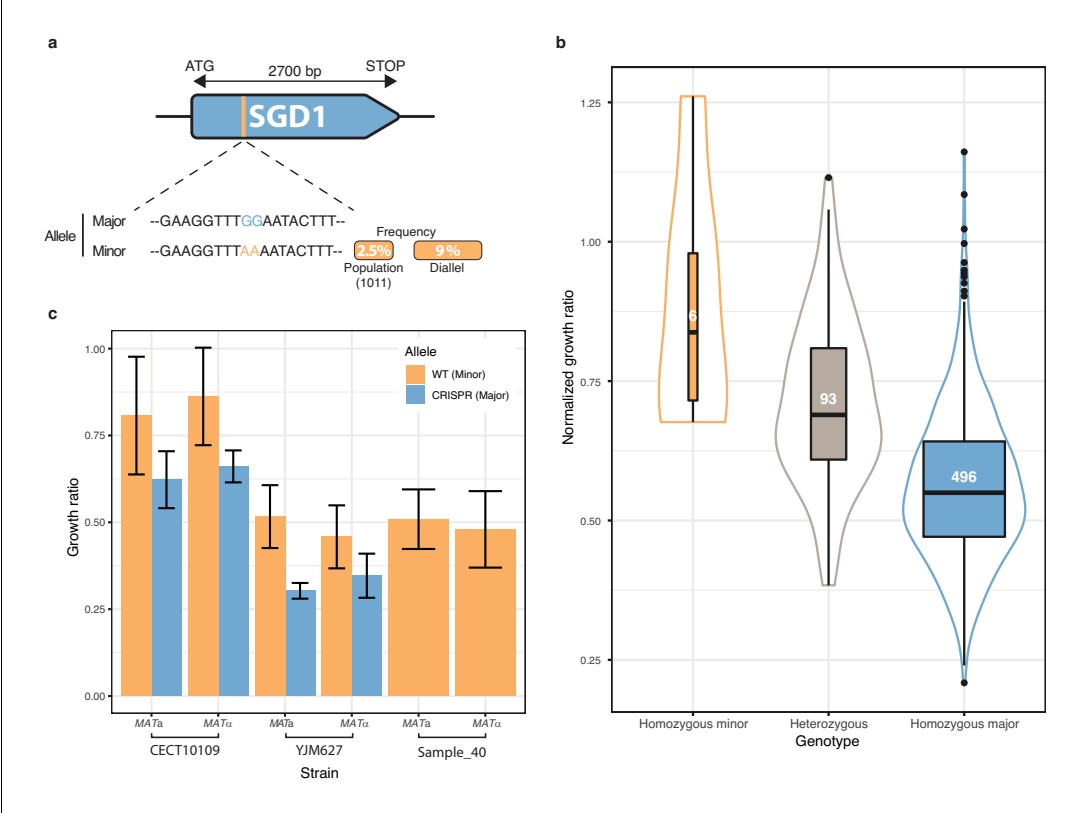

**Figure 5.** Low-frequency variant functional validation in 6-azauracil 100 µg.ml⁻¹. (a) Schematic representation of *SGD1* with the relative position of the detected SNPs. The minor allele is represented in orange with its MAF in the population and in the diallel cross panel. (b) boxplot and density plot of the normalized growth ratios for each genotype on 6-azauracil 100 µg.ml⁻¹. Number of observation is displayed in the boxplots. (c) Phenotypic validation after allele replacement of the minor allele with the major allele using CRISPR-Cas9 in the strains carrying the minor allele. Error bars represent median absolute deviation (four replicates).

DOI: https://doi.org/10.7554/eLife.49258.012

However, care should be taken with the classification of the mode of inheritance. Indeed, as we do not know how many loci are involved for each hybrid's phenotype, we can only assess the final phenotypic outcome of all the genetic variants involved and not on a locus by locus basis. This classification does not take into account their number, effect size and interactions. Consequently, the mode of inheritance that we described here solely reflects how the phenotype of the hybrid varies with respect to its parents. For example, several interactions could take place with opposite effect, leading to a final phenotype that appears as being controlled by an additive mode of inheritance (*i. e.* the hybrid phenotype equal to the mid parent value). However, in the cases where dominance was detected as a mode of inheritance, this might reflect the presence of a single locus having a strong phenotypic impact acting dominantly thus being responsible by itself for the phenotype. Yet, if two hybrids show a complete dominance in the same condition, it does not mean that the same alleles are involved in both.

Although few low-frequency and rare variants were considered in our GWAS (4%) due to stringent filtering conditions, a strong enrichment in these variants has been observed in the significantly associated ones (16%), demonstrating the ubiquity of low-frequency variants with important phenotypic impact. However, when looking at the population level, even though they do have effect sizes similar to common variants, they are not going to explain an important part of variance because it relies both on effect size and allele frequency. A good example of this phenomenon has been seen with a study of human height in more than 700,000 individuals. A total of 83 significantly associated rare and low-frequency variants with effect sizes up to 2 cm have been mapped (*Marouli et al., 2017*). On average, they explained the same amount of phenotypic variation as common variants,

which displayed much smaller effect sizes of about 1 mm. Our results suggest that a high number of low-frequency variants play a decisive role in the phenotypic landscape of a population both in term of number and effect size. Taken one by one, they do not explain a lot of phenotypic variance in a large population. Yet, altogether, they might actually explain a greater part of the variation than the one explained by common variants.

The contribution of rare and low-frequency variants to traits is largely unexplored. In humans, these genetic variants are widespread but only a few of them have been associated with specific traits and diseases (*Walter et al., 2015*). Recently, it has been shown that the missing heritability of height and body mass index is accounted for by rare variants (*Wainschtein et al., 2019*). We also recently found in yeast that most of the previously identified Quantitative Trait Nucleotides (QTNs) using linkage mapping were at low allele frequency in the 1,011 *S. cerevisiae* population (*Hou et al., 2016*; *Hou et al., 2019*; *Peltier et al., 2019*; *Peter et al., 2018*). A total of 284 QTNs were identified by linkage mapping and 150 of them are present at a low frequency in the population of 1011 isolates (*Peltier et al., 2019*; *Peter et al., 2018*). However, these QTNs were mapped with mostly closely related genetic backgrounds, encompassing a total of 59 strains with 30% of them coming from laboratory and 41% coming from the wine cluster, which has a very low genetic diversity (*Peter et al., 2018*). Moreover, experimentally validated QTNs are, most of the time, genetic variants with the most important phenotypic impact, which has been previously recognized as inducing an ascertainment bias (*Rockman, 2012*). It also raised the question of whether these rare and large effect size alleles discovered in specific crosses are really relevant to the variation across most of the population.

Here, we quantified the contribution of low-frequency variants across a large number of growth conditions and found that among all the genetic variants detected by GWAS on a diallel panel, 16.3% of them have a low-frequency in the initial population and explain a significant part of the phenotypic variance (21% on average). This particular diallel design also presents an intrinsic power to evaluate the additive vs. non-additive genetic components contributing to the phenotypic variation. We assessed the effect of intra-locus dominance on the non-additive genetic component and showed that dominance at the single locus level contributed to the phenotypic variation observed. However, other more complicated inter-loci interactions may still be involved. Altogether, these results have major implications for our understanding of the genetic architecture of traits in the context of unexplained heritability. In parallel to a recent large-scale linkage mapping survey in yeast (*Bloom et al., 2019*), our study highlights the extensive role of low-frequency variants on the phenotypic variation.

## Materials and methods

### Construction of the diallel panel

#### Selection of the *S. cerevisiae* isolates

Out of the collection of 1011 strains (*Peter et al., 2018*), a total of 53 natural isolates were carefully selected to be representative of the *S. cerevisiae* species. We selected isolates from a broad ecological origins and we prioritized for strains that were diploid, homozygous, euploid and genetically as diverse as possible, that is up to 1% of sequence divergence. All the isolate details, including ecological and geographical origins, are listed in *Supplementary file 1*. In addition to these 53 isolates, we included two laboratory strains, namely $\sum$1278b and the reference S288c strain.

### Generation of stable haploids

For each selected parental strain, stable haploid strains were obtained by deleting the *HO* locus. The *HO* deletions were performed using PCR fragments containing drug resistance markers flanked by homology regions up and down stream of the *HO* locus, using standard yeast transformation method. Two resistance cassettes, *KanMX* and *NatMX*, were used for *MAT*a and *MAT*α haploids, respectively. The mating-type (*MAT*a and *MAT*α) of antibiotic-resistant clones was determined using testers of well-known mating type. For each genetic background, we selected a *MAT*a and *MAT*α clone that are resistant to G418 or nourseothricin, respectively.

Phenotyping of the parental haploid strains was performed to check for mating type-specific fitness effects. All *MAT*a and *MAT*α parental strains were tested on all 49 growth conditions (see

below) using the same procedure as the phenotyping assay of the hybrid matrix. The overall correlation between the *MAT*a and *MAT*α parental strains was 0.967 (Pearson, p-value<1e-324), with an average correlation per strain of 0.976 across different conditions (*Figure 1—figure supplement 3*). No significant mating type specificity was identified.

## Diallel scheme

Parental strains were arrayed and pregrown in liquid YPD (1% yeast extract, 2% peptone and 2% glucose) overnight. Mating was performed with ROTOR (Singer Instruments) by pinning and mixing *MAT*a over *MAT*α parental strains on solid YPD. The parental strains, that is 55 *MAT*a *HO::ΔKanMX* and 55 *MAT*α *HO::ΔNatMX* strains were arrayed and mated in a pairwise manner on YPD for 24 hr at 30°C. The mating mixtures were replicated on YPD supplemented with G418 (200 µg.ml$^{-1}$) and nourseothricin (100 µg.ml$^{-1}$) for double selection of hybrid individuals. After 24 hr, plates were replicated again on the same media to eliminate potential residuals of non-hybrids cells. In total, we generated 3025 hybrids, representing 2970 heterozygous hybrids with a unique parental combination and 55 homozygous hybrids.

## High-throughput phenotyping and growth quantification

Quantitative phenotyping was performed using endpoint colony growth on solid media. Strains were pregrown in liquid YPD medium and pinned onto a solid SC (Yeast Nitrogen Base with ammonium sulfate 6.7 g.l$^{-1}$, amino acid mixture 2 g.l$^{-1}$, agar 20 g.l$^{-1}$, glucose 20 g.l$^{-1}$) matrix plate to a 1536 density format using the replicating ROTOR robot (Singer Instruments). Two biological replicates (coming from independent cultures) of each parental haploid strain were present on every plate and six biological replicates were present for each hybrid. As 27 plates were used in order to phenotype all the hybrids, 27 technical replicates (same culture in different plates) of the parents were present. The resulting matrix plates were incubated overnight to allow sufficient growth, which were then replicated onto 49 media conditions, plus SC as a pinning control (*Figure 1—figure supplement 1*, *Supplementary file 2*). The selected conditions impact a broad range of cellular responses, and multiple concentrations were tested for each compound (*Figure 1—figure supplement 2*). Most tested conditions displayed distinctive phenotypic patterns, suggesting different genetic basis for each of them (*Figure 1—figure supplement 2*). The plates were incubated for 24 hr at 30°C (except for 14°C phenotyping) and were scanned with a resolution of 600 dpi at 16-bit grayscale. Quantification of the colony size was performed using the R package Gitter (*Wagih and Parts, 2014*) and the fitness of each strain on the corresponding condition was measured by calculating the normalized growth ratio between the colony size on a condition and the colony size on SC. As each hybrid is present in six replicates, the value considered for its phenotype is the median of all its replicates, thus smoothing the effects of pinning defect or contamination. This phenotyping step led to the determination of 148,225 hybrid/trait combinations (*Figure 1—source data 1*).

## Diallel combining abilities and heritabilities

Combining ability values were calculated using half diallel with unique parental combinations, excluding homozygous hybrids from identical parental strains. For each hybrid individual, the fitness value is expressed using Griffing's model (*Griffing, 1956*):

$$z_{ij} = \mu + g_i + g_j + s_{ij} + e$$

Where $z_{ij}$ is the fitness value of the hybrid resulting from the combination of $i^{th}$ and $j^{th}$ parental strains, $\mu$ is the mean population fitness, $g_i$ and $g_j$ are the general combining ability for the $i^{th}$ and $j^{th}$ parental strains, $s_{ij}$ is the specific combining ability associated with the $i \times j$ hybrid, and $e$ is the error term ($i = 1...N$, $j = 1...N$, $N = 55$). General combining ability for the $i^{th}$ parent is calculated as:

$$g_i = \left(\frac{N-1}{N-2}\right) \times (\bar{z}_{i\cdot} - \mu)$$

Where $N$ is the total number of parental types, $\bar{z}_{i\cdot}$ is the mean fitness value of all half sibling hybrids involving the $i^{th}$ parent, and $\mu$ is the population mean. The error term associated with $g_i$ is:

$$e_{g_i} = \sqrt{\frac{(N-1) \times \sigma^2 z_{ij.}}{n \times N \times (N-2)}}$$

Where $N$ is the total number of parental types, $n$ is the number of replicates for the $i \times j$ hybrid, and $\sigma^2 z_{ij.}$ is the variance of fitness values from a full-sib family involving the $i^{th}$ and $j^{th}$ parents, which is expressed as:

$$\sigma^2 z_{ij.} = \sigma^2 z_i + \sigma^2 z_j + \sigma^2 z_{ij} + 2 \times cov(z_i, z_j)$$

Specific combining ability for the $i \times j$ hybrid combination therefore:

$$s_{ij} = \bar{z_{ij.}} - g_i - g_j - \mu$$

The error term associated with $s_{ij}$ is:

$$e_{s_{ij}} = \sqrt{\frac{(N-3) \times \sigma^2 z_{ij.}}{n \times (N-1)}}$$

Using combining ability estimates, broad- and narrow-sense heritabilities can be calculated. Narrow sense heritability ($h^2$) accounts for the part of phenotypic variance explained only by additive variance, expressed as the additive variance ($\sigma_A^2$) over the total phenotypic variance observed ($\sigma_P^2$):

$$h^2 = \frac{\sigma_A^2}{\sigma_P^2} = \frac{\sigma_{(g_i+g_j)}^2}{\sigma_{(g_i+g_j)}^2 + \sigma_{s_{ij}}^2 + \sigma_e^2}$$

Where $\sigma_{(g_i+g_j)}^2$ is the sum of GCA variances, $\sigma_{s_{ij}}^2$ is the SCA variance and $\sigma_e^2$ is the variance due to measurement error, which is expressed as:

$$\sigma_e^2 = (N-2)\left(\bar{e_{g_i}} + \bar{e_{g_j}}\right)^2 + \frac{\left(\frac{(N^2-N)}{2} - 1\right)}{\left(\frac{(N^2-N)}{2} + N - 3\right)} \times \bar{e_{s_{ij}}}^2$$

On the other hand, broad-sense heritability ($H^2$) depicts the part of the phenotypic variance explained by the total genetic variance $\sigma_G^2$:

$$H^2 = \frac{\sigma_G^2}{\sigma_P^2} = \frac{\sigma_{(g_i+g_j)}^2 + \sigma_{s_{ij}}^2}{\sigma_{(g_i+g_j)}^2 + \sigma_{s_{ij}}^2 + \sigma_e^2}$$

Phenotypic variance explained by non-additive variance is therefore equal to the difference between $H^2$ and $h^2$. All calculations were performed in R using custom scripts.

## Calculation of mid-parent values and classification of mode of inheritance

Mid-Parent Value (MPV) is expressed as the mean fitness value of both diploid homozygous parental phenotypes:

$$MPV = \frac{P1 + P2}{2}$$

Comparing the hybrid phenotypic value (*Hyb*) to its respective parents' allows for an inference of the mode of inheritance for each hybrid/trait combination (*Figure 3a–b*). To obtain a robust classification, confidence intervals for each class were based on the standard deviation of hybrid (six replicates) and parents (54 replicates). *P2* is the phenotypic value of the fittest parent while *P1* is the phenotypic value of the least fit parent.

| Inheritance mode | Formula |
|---|---|
| Underdominance | $Hyb1 - (\sigma_{P1} + \sigma_{Hyb})$ |
| Dominance P1 | $P1 - (\sigma_{P1} + \sigma_{Hyb})1 + (\sigma_{P1} + \sigma_{Hyb})$ |
| Partial dominance P1 | $P1 + (\sigma_{P1} + \sigma_{Hyb}) - (\frac{\sigma_{P1} + \sigma_{P2}}{2} + \sigma_{Hyb})$ |
| Additivity | $MPV + (\frac{\sigma_{P1} + \sigma_{P2}}{2} + \sigma_{Hyb})2 - (\sigma_{P2} + \sigma_{Hyb})$ |
| Partial dominance P2 | $MPV - (\frac{\sigma_{P1} + \sigma_{P2}}{2} + \sigma_{Hyb}) + (\frac{\sigma_{P1} + \sigma_{P2}}{2} + \sigma_{Hyb})$ |
| Dominance P2 | $P2 - (\sigma_{P2} + \sigma_{Hyb})2 + (\sigma_{P2} + \sigma_{Hyb})$ |
| Overdominance | $P2 + (\sigma_{P2} + \sigma_{Hyb})$ |

When a clear separation is possible between the two parental phenotypic values $(P1 + \sigma_{P1}2 - \sigma_{P2})$, the full decomposition in the seven above mentioned categories is possible (*Figure 3a*). However, in most of the cases, the two parental phenotypic values are not separated enough to achieve this but it is still possible to distinguish between overdominance and underdominance (*Figure 3b*, *Figure 3d*). All calculations were performed in R using custom scripts.

## Genome-wide association studies on the diallel panel

Whole genome sequences for the parental strains were obtained from the 1002 yeast genome project (*Peter et al., 2018*). Sequencing was performed by Illumina Hiseq 2000 with 102 bases read length. Reads were then mapped to S288c reference genome using bwa (v0.7.4-r385) (*Li and Durbin, 2009*). Local realignment around indels and variant calling has been performed with GATK (v3.3–0) (*McKenna et al., 2010*). The genotypes of the F1 hybrids were constructed in silico using 34 parental genome sequences. We retained only the biallelic polymorphic sites, resulting in a matrix containing 295,346 polymorphic sites encoded using the 'recode12' function in PLINK (*Chang et al., 2015*). Those genotypes correspond to a half-matrix of pairwise crosses with unique parental combinations, including the diagonal, that is the 34 homozygous parental genotypes. For each cross, we combined the genotypes of both parents to generate the hybrid diploid genome. As a result, heterozygous sites correspond to sites for which the two parents had different allelic versions. We removed long-range linkage disequilibrium sites in the diallel matrix due to the low number of founder parental genotypes by removing haplotype blocks that are shared more than twice across the population, resulting in a final dataset containing 31,632 polymorphic sites.

We performed GWA analyses with different encodings (*Seymour et al., 2016*). In the additive model, the genotypes of the F1 progeny were simply the concatenation of the genotypes from the parents. As homozygous parental alleles were encoded as 1 or 2, the possible alleles for each site in the F1 genotype were '11' and '22' for homozygous sites and '12' for heterozygous sites. We also used an overdominant genotype encoding, where both the homozygous minor and homozygous major alleles were encoded as '11' and the heterozygous genotype was encoded as '22'.

Mixed-model association analysis was performed using the FaST-LMM python library version 0.2.32 (https://github.com/MicrosoftGenomics/FaST-LMM) (*Widmer et al., 2015*). We used the normalized phenotypes by replacing the observed value by the corresponding quantile from a standard normal distribution, as FaST-LMM expects normally distributed phenotypes. The command used for association testing was the following: single_snp(bedFiles, pheno_fn, count_A1 = True), where bedFiles is the path to the PLINK formatted SNP data and pheno_fn is the PLINK formatted phenotype file. By default, for each SNP tested, this method excludes the chromosome in which the SNP is found from the analysis in order to avoid proximal contamination. Fast-LMM also computes the fraction of heritability explained for each SNP. The mixed model adds a polygenic term to the standard linear regression designed to circumvent the effects of relatedness and population stratification.

We estimated a condition-specific p-value threshold for each condition by permuting phenotypic values between individuals 100 times. The significance threshold was the 5% quantile (the 5th lowest p-value from the permutations). With that method, variants passing this threshold will have a 5% family-wise error rate. However, we do not have any estimation of the false positive rate. Taken together, GWA revealed 1723 significantly associated SNPs (*Figure 4—source data 1*), with 1273 and 450 SNPs for overdominant and additive model, respectively.

## Variance explained and effect size

Variance explained by each SNP is calculated by PLINK. Care must be taken that in order to obtain the variance explained by all SNPs, it is not possible to sum up the variance explained by each individual SNP based on the fact that SNPs are not completely independent from one another.

The effect size was calculated using the formula for Cohen's d:

$$d = \frac{\bar{x_1} - \bar{x_2}}{sd_{Pooled}}$$

Where the pooled standard deviation is calculated with the following formula:

$$sd_{Pooled} = \sqrt{\frac{sd_1^2 + sd_2^2}{2}}$$

Under the additive model, the heterozygote phenotype is equidistant to both possible homozygote phenotypes (minor allele and major allele), so our calculation of the effect size could either compare the heterozygotes with the homozygotes in the minor allele, or the heterozygotes with the homozygotes in the major alleles. We chose to use the latter since the major allele grants us more statistical power. The formula we used to obtain the effect size for a given SNP under this model is the following:

$$Effect\ size = \frac{\bar{x}_{Heterozygous} - \bar{x}_{Major}}{sd_{Pooled}}$$

Under the overdominant model, the heterozygote phenotype is compared to the phenotype of the group of both homozygotes (minor and major), so the formula we used to obtain the effect size for a given SNP under this model is the following:

$$Effect\ size = \frac{\bar{x}_{Heterozygous} - \bar{x}_{Major\ and\ Minor}}{sd_{Pooled}}$$

## Gene ontology analysis

GO term enrichment was performed using SGD GO Term Finder (https://www.yeastgenome.org/goTermFinder) with the 546 unique genes containing significantly associated SNPs (*Figure 4—source data 1* and *Supplementary file 3*). Significant enrichment is considered under 'Process' ontology with a p-value cutoff of 0.05.

## CRISPR-Cas9 allele editing

pAEF5 plasmid containing Cas9 endonuclease and the guide RNA targeting *SGD1* was co-transformed with the repair fragment of 100 nucleotides containing the desired allele. Transformed cells were then plated on YPD supplemented with 200 µg.ml$^{-1}$ hygromycin at 30°C to select for transformants. Colonies were then arrayed on a 96 well plate with 100 µl YPD and grown for 24 hr to induce plasmid loss. The plate was then pinned back onto solid YPD for 24 hr then replica plated to YPD supplemented with 200 µg.ml$^{-1}$ hygromycin to check for plasmid loss. Allele specific PCR was performed on colonies that lost the plasmid (*Wangkumhang et al., 2007*) to distinguish correctly edited allele from wildtype allele. Strains who showed amplification for the edited allele and no amplification for the wildtype allele were phenotyped (four technical replicates and four biological replicates) on the corresponding condition to measure differences with their wildtype counterparts.

## Statistical tests

Person's correlation test was used to assess linear correlation between two sets.

Wilcoxon Mann Whitney was used to determine if two independent samples have the same distribution.

Correlogram of all tested growth conditions. Numbers in each cell represent 100 x Pearson's r value.

## Acknowledgements

We thank Joshua Bloom and Leonid Kruglyak for insightful discussions, comments on the manuscript as well as for sharing their unpublished manuscript. We thank Maitreya Dunham and the members of the Schacherer laboratory for comments and suggestions. We also thank Gilles Fischer for providing the pAEF5 plasmid. This work was supported by a National Institutes of Health (NIH) grant R01 (GM101091-01) and a European Research Council (ERC) Consolidator grant (772505). TF is supported in part by a grant from the Ministère de l'Enseignement Supérieur et de la Recherche and in part by a fellowship from the medical association la Fondation pour la Recherche Médicale. JS is a Fellow of the University of Strasbourg Institute for Advanced Study (USIAS) and a member of the Institut Universitaire de France.

## Additional information

### Funding

| Funder | Grant reference number | Author |
|---|---|---|
| National Institutes of Health | R01 GM101091-01 | Joseph Schacherer |
| European Research Council | Consolidator grants (772505) | Joseph Schacherer |
| Fondation pour la Recherche Médicale | Graduate student grant | Téo Fournier |
| Institut Universitaire de France | | Joseph Schacherer |
| University of Strasbourg Institute for Advanced Study | | Joseph Schacherer |
| Ministère de l'Enseignement Supérieur et de la Recherche | | Téo Fournier |

The funders had no role in study design, data collection and interpretation, or the decision to submit the work for publication.

### Author contributions

Téo Fournier, Conceptualization, Resources, Data curation, Software, Formal analysis, Investigation, Visualization, Methodology, Writing—original draft, Writing—review and editing; Omar Abou Saada, Software, Formal analysis, Writing—review and editing; Jing Hou, Conceptualization, Software, Formal analysis, Methodology, Writing—review and editing; Jackson Peter, Software, Formal analysis; Elodie Caudal, Resources, Investigation, Writing—review and editing; Joseph Schacherer, Conceptualization, Supervision, Funding acquisition, Validation, Methodology, Writing—original draft, Project administration, Writing—review and editing

### Author ORCIDs

Téo Fournier ⬛ https://orcid.org/0000-0002-4860-6728
Joseph Schacherer ⬛ https://orcid.org/0000-0002-6606-6884

### Decision letter and Author response

Decision letter https://doi.org/10.7554/eLife.49258.020
Author response https://doi.org/10.7554/eLife.49258.021

## Additional files

### Supplementary files

• Supplementary file 1. Strains used for the diallel cross with their ecological and geographical origins.
DOI: https://doi.org/10.7554/eLife.49258.013
• Supplementary file 2. Phenotyping conditions and their respective type of induced stress.

DOI: https://doi.org/10.7554/eLife.49258.014

• Supplementary file 3. GO Term associated with the 546 unique genes with a significantly associated SNPs.
DOI: https://doi.org/10.7554/eLife.49258.015

• Transparent reporting form DOI: https://doi.org/10.7554/eLife.49258.016

## Data availability

All data generated or analysed during this study are included in the manuscript and supporting files. Source data files have been provided for Figures 1 and 4.

The following previously published dataset was used:

| Author(s) | Year | Dataset title | Dataset URL | Database and Identifier |
|---|---|---|---|---|
| Jackson Peter, Matteo De Chiara, Anne Friedrich, Jia-Xing Yue, David Pflieger, Anders Bergström, Anastasie Sigwalt, Benjamin Barre, Kelle Freel, Agnès Llored, Corinne Cruaud, Karine Labadie, Jean-Marc Aury, Benjamin Istace, Kevin Lebrigand, Pascal Barbry, Stefan Engelen, Arnaud Lemainque, Patrick Wincker, Gianni Liti, Joseph Schacherer | 2018 | Genome evolution across 1,011 Saccharomyces cerevisiae isolates | https://www.ncbi.nlm.nih.gov/sra?term=ERP014555 | NCBI SRA, ERP014555 |

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
