## [Decision Letter]

**Acceptance summary:**

The authors examine the relationship between the frequency of genetic variants in natural populations and their effects on complex growth traits using the budding yeast as a model. They find that high-impact variants tend to be rare and that their effects often combine in a non-additive manner. Their results contribute to a better understanding of phenotypic diversity and will help future developments in the use of natural populations for the mapping of genetic variation underlying complex traits such as those using GWAS in which low-frequency variants represent a particular challenge. Their observations are therefore of interest to a large community of scientists interested in evolution, genetics and particularly in the architecture of complex traits. The data produced and approach developed also represent an important resource for the community.

**Decision letter after peer review:**

Thank you for submitting your article "Extensive impact of low-frequency variants on the phenotypic landscape at population-scale" for consideration by *eLife*. Your article has been reviewed by three peer reviewers, and the evaluation has been overseen by a Reviewing Editor and Naama Barkai as the Senior Editor. The reviewers have opted to remain anonymous.

The reviewers have discussed the reviews with one another and the Reviewing Editor has drafted this decision to help you prepare a revised submission.

Summary:

Your paper examines the correlation between allele frequencies and their effects on quantitative characters using QTL mapping and the analysis of a large number of genomes. You find that rare variants explain an unexpectedly large proportion of phenotypic variance. Your study is one of the first to examine this association systematically. Overall, the reviewers found the work of interest and to be a potentially important contribution. One major concern that emerged from the reviews and the discussions among the reviewers is that the importance of the work will not be obvious for non-specialists. One reviewer also mentions that similar conclusions could have been obtained from a meta-analysis of the existing literature. Since *eLife* is a generalist journal, it would be crucial to better articulate why the study is important and how the findings will impact the field of genetics and maybe evolution in general. More theoretical background as to why variants with large impacts on phenotypes should be rare or vice-versa would be useful. The manuscript is currently very short so you have plenty of space to extend on these points in the Introduction and in the Discussion. One reviewer also suggested you extend the analysis and text on the implication of the conditions tested for yeast biology, which I believe would strengthen the paper as well in terms of impact.

I collated below the other comments of the reviewers that are essential points to consider if you want to submit a revised version.

Essential revisions:

1) For a polygenic trait, the distinction between dominance and additivity isn't a relevant one. For example, you could have 100 loci, each is completely dominance, but if they are additive between loci, the hybrid test will appear additive. The latter results by GWAS suggest that a lot of variants have over-dominant effect (at least some over-dominant component). I can see what the authors are trying to do here, i.e., to assess the contribution of additivity versus other non-additive effects, but I think as long as there are many loci and there is some degree of additivity between loci, everything will appear additive. I think the distinction between additive and non-additive effects are only relevant when discussing one locus. If you had a panel of near-isogenic lines, a diallel experiment could answer the question of additivity versus non-additivity. The results from this analysis are still useful and I would suggest the authors simply report the results without invoking the term of additivity versus dominance. Alternatively, clearly state the caveats so readers don't mis-read the interpretation.

2) I have a somewhat different interpretation of the rare versus common comparison. There are a few facts nicely presented. 1) although there are fewer rare variants in the diallel than common ones, rare variants are more likely to be associated with the traits. This is a major finding. 2) On a per variant basis, common and low-frequency variants explain about the same amount of variation. This means the effect size should be larger for rare variants than common variants. I don't think the statistical significance in Figure 4D is worth highlighting, the difference was minimal (20.2% versus 19.6% with a large variance). Power is proportional to variance explained so it's expected that these two groups produce more or less equal variance on a per variant basis if using the same threshold. However, in the diallel, there are way more common variants than rare variants. This means in the diallel, more variance is explained by common variants as a whole. I can see that if rare variants are more likely to be associated with traits, then in an outbred population, they could also be disproportionally associated with traits but more difficult to detect. I would appreciate some discussion on the contribution by a per-variant basis and overall contribution.

3) The main conclusion of the manuscript is that rare variants significantly contribute to genetic variance. In my view, this conclusion is biased as these rare causal variants are being analyzed in genetic backgrounds in which they are no longer rare; actually, these variants are biallelic. Several studies have shown that a rare variant of MKT1(89A) is a significant contributor to phenotypic variation whenever it is present in segregating populations. However, MKT1(89A) allele hardly identified when one of the parents is not S288c, the strain which harbours this allele. So the extension that if the rare variant has a significant effect in a sub-population, its effect size would be similar in a large heterogeneous population is false. Furthermore, the authors conclude that their larger 55 strain population, a representative distribution of 1000 strain collection, most of the variants have additive effects. This the authors claim is revalidation of other previous studies (Bloom et al., 2013, 2015), where they identified most of the causal variants between BYxRM had additive effects. However, subsequent papers (Frosberg et al. 2017, PMID 28250458; Yadav et al. 2016) showed that variance mapping in BYxRM segregants helped to account for genetic interactions and showed how non-additive interactions also contribute significantly to phenotypic variation. One of the results in the manuscript that non-additive effects contribute 1/3rd to phenotypic variance indicates that additive effects do not explain all effects with dominance, a non-additive interaction, being a significant contributor. Also, the authors fail to explain why dominance is so frequently observed in their diallelic panel. A possible reason could be that one variant is selected for a trait better than the other, and in combination with a weaker or neutral allele, it shows dominance.

4) I find that just doing a few more strains does not make this manuscript a significant advance over the previous studies. One can argue that taking into account all causal variants identified to date (Fay, 2013), one can identify what frequency of rare variants have been identified, e.g. a typical example being MKT1(89A) allele as causal, even though their effect size will not be identified using this strategy. Peltier et al., 2019, show that 284 rare QTNs variants have been identified to date and these functional variants being private to a subpopulation, possibly due to their adaptive role to a specific environment. Moreover, this conclusion can be made without these extensive experimental crosses.

---

## [Author Response]

Your paper examines the correlation between allele frequencies and their effects on quantitative characters using QTL mapping and the analysis of a large number of genomes. You find that rare variants explain an unexpectedly large proportion of phenotypic variance. Your study is one of the first to examine this association systematically. Overall, the reviewers found the work of interest and to be a potentially important contribution. One major concern that emerged from the reviews and the discussions among the reviewers is that the importance of the work will not be obvious for non-specialists. One reviewer also mentions that similar conclusions could have been obtained from a meta-analysis of the existing literature.

We performed such an analysis in the framework of the 1002 Yeast Genomes Project and this analysis was mentioned in the first version of the manuscript. More recently, we were involved in a larger analysis but this one was not cited (Peltier et al., 2019) because unpublished at that time. Now, a proper citation has been included and we commented on this specific point in the Discussion.

Even if such analyses are really insightful, we really think that there are some biases in the subset of detected QTNs in yeast using linkage mapping for different reasons: First in terms of genetic backgrounds studied as most of linkage mapping studies were performed on mostly the same set of isolates. Second, experimentally validated QTNs are often prioritized based on their effect size.

Our study allows for a more global and quantitative approach as the variants are taken from a representative, genetically diverse and larger population. The subset of genetic variants is also much larger. Overall, this dataset gives a precise as well as a quantitative global view of the role of low-frequency variants on the phenotypic diversity in a population.

Since eLife is a generalist journal, it would be crucial to better articulate why the study is important and how the findings will impact the field of genetics and maybe evolution in general. More theoretical background as to why variants with large impacts on phenotypes should be rare or vice-versa would be useful. The manuscript is currently very short so you have plenty of space to extend on these points in the Introduction and in the Discussion.

As suggested, we modified the Introduction by adding more background on the missing heritability problem as well as on the role of low-frequency and rare variants in human diseases. We also expanded the Discussion in order to answer to several points raised during the reviewing process (see below).

One reviewer also suggested you extend the analysis and text on the implication of the conditions tested for yeast biology, which I believe would strengthen the paper as well in terms of impact.

The goal of our study was to have a myriad of complex traits to study. Consequently we selected a large number of conditions for which the phenotypic variance was broad in our population. These conditions were already tested in the framework of the 1002 Yeast Genomes Project (Peter et al., 2018). Most of them show a normal distribution, meaning that they correspond to complex traits. A good dissection and analysis of the implication of the tested conditions for yeast biology require an additional step, namely the determination of inheritance patterns in the progeny. This is actually something that is intended as a logical follow-up to this study.

Essential revisions:1) For a polygenic trait, the distinction between dominance and additivity isn't a relevant one. For example, you could have 100 loci, each is completely dominance, but if they are additive between loci, the hybrid test will appear additive. The latter results by GWAS suggest that a lot of variants have over-dominant effect (at least some over-dominant component). I can see what the authors are trying to do here, i.e., to assess the contribution of additivity versus other non-additive effects, but I think as long as there are many loci and there is some degree of additivity between loci, everything will appear additive. I think the distinction between additive and non-additive effects are only relevant when discussing one locus. If you had a panel of near-isogenic lines, a diallel experiment could answer the question of additivity versus non-additivity. The results from this analysis are still useful and I would suggest the authors simply report the results without invoking the term of additivity versus dominance. Alternatively, clearly state the caveats so readers don't mis-read the interpretation.

As we only look at the final phenotype of the hybrid, we do agree that the distinction of additivity vs. dominance is only the result of all the combined effects of the genes and that no distinction between the effect of individual loci can be done. However, one can argue that if dominance is indeed detected as the main mode of inheritance, it might suggest the presence of a locus of high phenotypic impact acting dominantly. Also it is possible that if two hybrids display complete dominance towards a parent, it does not necessarily reflect that the same locus is involved in both cases. As suggested, we clearly stated the caveats and consequently we added a paragraph in the Discussion to clarify this point.

2) I have a somewhat different interpretation of the rare versus common comparison. There are a few facts nicely presented.1) although there are fewer rare variants in the diallel than common ones, rare variants are more likely to be associated with the traits. This is a major finding.

We thank the reviewer for this comment. It is, indeed, true that low-frequency variants are disproportionally associated to the trait (i.e. they are overrepresented) and we now emphasized more on that point in the Abstract and the Results section.

2) On a per variant basis, common and low frequency variants explain about the same amount of variation. This means the effect size should be larger for rare variants than common variants. I don't think the statistical significance in Figure 4D is worth highlighting, the difference was minimal (20.2% versus 19.6% with a large variance). Power is proportional to variance explained so it's expected that these two groups produce more or less equal variance on a per variant basis if using the same threshold. However, in the diallel, there are way more common variants than rare variants. This means in the diallel, more variance is explained by common variants as a whole. I can see that if rare variants are more likely to be associated with traits, then in an outbred population, they could also be disproportionally associated with traits but more difficult to detect. I would appreciate some discussion on the contribution by a per-variant basis and overall contribution.

We thank the reviewer for these comments. This is only true if we look at it in the same population. However, here, in our diallel panel, the low-frequency variants in the initial population are no longer rare because of a shift of the allele frequency. For example, a variant having a MAF of 3% in the 1,011 can rise to 25% in the diallel. Thus, the fraction explained in the diallel won’t be linked to the MAF in the initial population.

To answer this issue, we computed the effect size of the significantly associated variants. Effect size is a metric that is independent of allele frequency thus making it more prone to extrapolation in a different population. We added a paragraph about this point in the Results section as well as a figure (Figure 3E), and in the Discussion.

Concerning the fraction explain by common and low frequency associated SNPs, we do agree that the difference is minimal. As suggested, we did not highlight that point in the new version anymore.

3) The main conclusion of the manuscript is that rare variants significantly contribute to genetic variance. In my view, this conclusion is biased as these rare causal variants are being analyzed in genetic backgrounds in which they are no longer rare; actually, these variants are biallelic. Several studies have shown that a rare variant of MKT1(89A) is a significant contributor to phenotypic variation whenever it is present in segregating populations. However, MKT1(89A) allele hardly identified when one of the parents is not S288c, the strain which harbours this allele. So the extension that if the rare variant has a significant effect in a sub-population, its effect size would be similar in a large heterogeneous population is false.

This part is related to what we mentioned previously. Indeed, the effect size of this variant would be roughly the same in a different population, however it is true that the fraction of variance explained by such a variant could be different. Consequently, we computed the effect size of the significantly associated variants and we’ve shown that effect size of low-frequency variants is not much different from common variants.

Furthermore, the authors conclude that their larger 55 strain population, a representative distribution of 1000 strain collection, most of the variants have additive effects. This the authors claim is revalidation of other previous studies (Bloom et al., 2013, 2015), where they identified most of the causal variants between BYxRM had additive effects. However, subsequent papers (Frosberg et al. 2017, PMID 28250458; Yadav et al. 2016) showed that variance mapping in BYxRM segregants helped to account for genetic interactions and showed how non-additive interactions also contribute significantly to phenotypic variation. One of the results in the manuscript that non-additive effects contribute 1/3rd to phenotypic variance indicates that additive effects do not explain all effects with dominance, a non-additive interaction, being a significant contributor. Also, the authors fail to explain why dominance is so frequently observed in their diallelic panel. A possible reason could be that one variant is selected for a trait better than the other, and in combination with a weaker or neutral allele, it shows dominance.

As suggested, we added the references in the text. One hypothesis that could be proposed to explain the importance of dominance in our dataset is the presence of genetic variants with strong phenotypic effect acting dominantly in some strains and being responsible for most of the phenotypic variance in all crosses being heterozygous at this particular locus. We now added this point in the Discussion section.

4) I find that just doing a few more strains does not make this manuscript a significant advance over the previous studies. One can argue that taking into account all causal variants identified to date (Fay, 2013), one can identify what frequency of rare variants have been identified, e.g. a typical example being MKT1(89A) allele as causal, even though their effect size will not be identified using this strategy. Peltier et al., 2019, show that 284 rare QTNs variants have been identified to date and these functional variants being private to a subpopulation, possibly due to their adaptive role to a specific environment. Moreover, this conclusion can be made without these extensive experimental crosses.

As already discussed above, we strongly believe that our study corresponds to a more global and systematic approach than the concatenation of different results from different linkage mapping studies. We exhaustively looked and compared the fraction of variance explained and the effect size from variants of a large dataset of associated genetic variants, which were not chosen based on their effect size.